# Genome-Wide Identification and Expression Analysis of the SQUAMOSA Promoter-Binding Protein-like (*SPL*) Transcription Factor Family in *Catalpa* *bungei*

**DOI:** 10.3390/ijms25010097

**Published:** 2023-12-20

**Authors:** Erqin Fan, Caixia Liu, Zhi Wang, Shanshan Wang, Wenjun Ma, Nan Lu, Yuhang Liu, Pengyue Fu, Rui Wang, Siyu Lv, Guanzheng Qu, Junhui Wang

**Affiliations:** 1State Key Laboratory of Tree Genetics and Breeding, Northeast Forestry University, Harbin 150040, China; fanerqin2012@hotmail.com (E.F.); liucaixia2020@outlook.com (C.L.); wss60392411@163.com (S.W.); lyh7321@nefu.edu.cn (Y.L.); fupengyue365@gmail.com (P.F.); sorry741786616@163.com (R.W.); 13089711696@163.com (S.L.); gzqu@nefu.edu.cn (G.Q.); 2State Key Laboratory of Tree Genetics and Breeding, Key Laboratory of Tree Breeding and Cultivation of National Forestry and Grassland Administration, National Innovation Alliance of Catalpa bungei, Research Institute of Forestry, Chinese Academy of Forestry, Beijing 100091, China; wangzhi6666@126.com (Z.W.); mwjlx.163@163.com (W.M.); ln_890110@163.com (N.L.); 3College of Life Science, Northeast Forestry University, Harbin 150040, China

**Keywords:** *Catalpa* *bungei*, *CbuSPL*, flowering

## Abstract

As a plant-specific transcription factor, the *SPL* gene family plays a critical role in plant growth and development. Although the *SPL* gene family has been identified in diverse plant species, there have been no genome-wide identification or systematic study reports on the *SPL* gene family in *Catalpa bungei*. In this study, we identified 19 putative *SPL* gene family members in the *C. bungei* genome. According to the phylogenetic relationship, they can be divided into eight groups, and the genes in the same group have a similar gene structure and conserved motifs. Synteny analysis showed that fragment duplication played an important role in the expansion of the *CbuSPL* gene family. At the same time, *CbuSPL* genes have cis-acting elements and functions related to light response, hormone response, growth and development, and stress response. Tissue-specific expression and developmental period-specific expression analysis showed that *CbuSPL* may be involved in flowering initiation and development, flowering transition, and leaf development. In addition, the ectopic expression of *CbuSPL4* in Arabidopsis confirmed that it can promote early flowering and induce the expression of related flowering genes. These systematic research results will lay a foundation for further study on the functional analysis of *SPL* genes in *C. bungei*.

## 1. Introduction

Gene expression plays a crucial role in plant growth and development, and transcription factors (TFs), a class of proteins that can bind to specific nucleotide sequences upstream of genes, can greatly affect gene expression. This gene family is a collection of multiple genes with similar structures and functions, and they play a specific role in organisms. At present, many plant-specific transcription factor families have been identified in plants, such as NAC (NAM, ATAF, and CUC) [1], AP2/ERF [2], and SPL (SQUAMOSA promoter-binding protein-like) [3].

The SQUAMOSA promoter-binding protein-like (*SPL*) genes form a major family of plant-specific transcription factors, mainly related to flower development. *SPL* genes were first identified to regulate the expression of MADS-box genes during early flower development in *Antirrhinum majus* [4]. The *SPL* gene family encodes a highly conserved SBP domain containing approximately 76 amino acid residues, including 2 tandem zinc fingers (Cys-Cys-His-Cys and Cys-Cys-Cys-His) and a nuclear localization signal (NLS) at the C-terminus [3,5,6].

Research shows that *SPL* genes are widely involved in plant growth and development processes, including the initiation of flowering [7,8], phase transition from juvenile to adult, vegetative growth to reproductive growth [9,10,11], floral organ development [12], fruit development [13], phytohormone signal transduction [14], and response to abiotic stress [15,16,17]. In Arabidopsis, *AthSPL3*, *SPL4*, and *SPL5* are closely related members, and the overexpression of all three genes promotes vegetative phase changes and flowering [10,18]. It was found that the Arabidopsis SPL protein can specifically bind to the conserved sequence motif in the promoter region of the *A. thaliana* floral meristem identity gene *AP1* [5,18]. Although *SPL* has been shown to be involved in a variety of biological processes, the functional study of *SPL* in *Catalpa bungei* is still very limited.

*Catalpa bungei* is a perennial woody plant belonging to the genus *Catalpa* (Bignoniaceae) [19,20]. It is an ancient tree species with ornamental, economic, and medicinal values unique to China [21,22,23]. It takes 5–7 years to bloom, which seriously limits the work of hybrid breeding and genetic improvement [20,24]. *Catalpa* ‘Bairihua’ is an excellent hybrid with multi-season flowering obtained through the hybrid breeding of *Catalpa bungei* ‘Luo Qiu 4′ and *C. fargesii* Bur. f. *duclouxii* (http://rif.caf.ac.cn/News.aspx?ItemID=6009, accessed on 20 June 2022) [25,26]. Moreover, it can bloom in the same year through asexual reproduction methods such as grafting, which is very rare in woody plants [25]. The acquisition of *Catalpa* ‘Bairihua’ provides excellent materials for the study of flowering development and hybrid breeding of *C. bungei* and other woody plants. Therefore, it is crucial to explore the molecular mechanisms of flowering time regulation and reproductive transition of *Catalpa* ‘Bairihua’ for its genetic improvement and utilization.

As a plant-specific transcription factor, the *SPL* gene family plays a critical role in plant growth and development. The *SPL* gene family has been identified in diverse plant species, such as *A. thaliana* [9,27], rice (*Oryza sativa*) [6], soybean (*Glycine max*) [28], maize (*Zea mays*) [29], tomato (*Solanum lycopersicum* L.) [30], grape (*Vitis vinifera*) [31], strawberry (*Fragaria vesca*) [32], poplar (*Populus trichocarpa*) [33], tea plant (*Camellia sinensis*) [34], apple (*Malus domestica* Borkh.) [35], *Ziziphus jujuba* [36], *Fraxinus mandshurica* [37], sweet cherry (*Prunus avium* L.) [13], and orchids [38]. However, there have been no genome-wide identification or systematic study reports on the *SPL* gene family in *C. bungei*. In this study, we identified 19 putative *SPL* gene family members in the *C. bungei* genome. Subsequently, chromosome localization, phylogenetic analysis, gene structure analysis, conserved motif analysis, cis-acting elements analysis, and synteny analysis were performed. In addition, the expression patterns of 19 *SPL* genes in different tissues of *C. bungei* and flower buds of *C. bungei* and *Catalpa* ‘Bairihua’ at different developmental stages were analyzed. Finally, the *CbuSPL4* gene was ectopically expressed in *Arabidopsis* to explore its gene function. Exploring the regulatory function of *SPL* gene family in the process of flowering and development of *C. bungei* is of great significance for the breeding and genetic improvement of *C. bungei*. The systematic research results will lay a foundation for further study on the functional analysis of *SPL* genes in *C. bungei*, and also provide new insights for further study on the biological function of *SPL* genes in the process of flowering transition and flower development of *Catalpa* ‘Bairihua’.

## 2. Results

### 2.1. Identification of CbuSPL Genes in C. bungei

To identify *SPL* genes in *C. bungei*, BLASTP analysis and HMM were performed against the whole genome sequence. A total of 19 putative *SPL* family genes identified in *C. bungei* were named from *CbuSPL1* to *CbuSPL19* according to their position from top to bottom on each corresponding chromosome and different chromosomes from chromosome 1 to chromosome 20 (Figure 1 and Appendix A). Among them, the six *CbuSPL* genes with the largest number were located on chromosome 18, followed by three genes on chromosomes 1 and 2 and two genes on chromosomes 13 and 15, respectively. There was only one gene on chromosomes 7, 10, and 17, respectively. The characteristics of these genes were analyzed and summarized in detail, including their gene length, CDS length, amino acid length, number of exons, Mw, pI, and subcellular localization prediction information. Their gene length ranged from 1180 to 6654 bp; the largest protein was encoded by *CbuSPL17* with 1088 amino acids and a CDS length of 3267 bp, whereas the smallest protein was encoded by *CbuSPL6* with 170 amino acids and a CDS length of 513 bp. The Mw of these proteins ranged from 18,894.06 (*CbuSPL6*) to 119,687.73 kDa (*CbuSPL17*) and pI varied from 6.12 (*CbuSPL14*) to 9.59 (*CbuSPL16*) (Table 1). In other analyses, we found significant differences in the number of exons ranging from 2 to 11 and the number of introns ranging from 1 to 10 for all *CbuSPL* genes; the mean number of exons was 4.47 (Table 1). Subcellular localization results showed that all CbuSPL proteins were localized in the nucleus (Table 1).

### 2.2. Sequence Alignments of CbuSPL Genes in C. bungei

The differences between the 19 CbuSPL proteins were analyzed through multiple-sequence alignment, and the results showed that all members of CbuSPL proteins contained a typical highly conserved SBP domain, which contained 74 amino acid residues, including two zinc finger motifs (Zn-1, Zn-2) and a nuclear localization signal (NLS) (Figure 2).

### 2.3. Phylogenetic Analysis of SPL Genes

To understand the phylogenetic relationships of the *SPL* family, we constructed a phylogenetic tree using all SPL full-length protein sequences from *C. bungei* (19 genes), *Arabidopsis thaliana* (17 genes), *Populus trichocarpa* (28 genes), *Malus domestica* (34 genes), *Sesamum indicum* (19 genes), *Solanum lycopersicum* (15 genes), and *Oryza sativa* (19 genes). A total of 151 SPL proteins from these seven species were classified into eight groups (I, II, III, VI, V, VI, VII, and VIII), each of which contained one or more CbuSPL proteins. The three largest groups have 40 (Group VIII), 28 (Group VII), and 18 (Group V) members, respectively (Figure 3).

### 2.4. Conserved Motifs and Gene Structure Analysis of CbuSPL Genes

In order to understand the differences in the gene structure of *CbuSPL* genes in *C. bungei*, we used TBtools v1.127 to visualize the gene structure of the *CbuSPL* genes, including the untranslated region (UTR), exons, and introns. A phylogenetic tree of 19 *CbuSPLs* was constructed to further analyze their evolutionary relationships (Figure 4a). To analyze the diversity and similarity of *CbuSPL* gene structures, 10 motifs were identified on the MEME website. The results show that all *CbuSPL* members contain more than three conserved motifs, including Motif 1, Motif 2, and Motif 3. Motif 8 only appears in *CbuSPL16* and *CbuSPL17* (in the same evolutionary branch). At the same time, *CbuSPL* genes with a close evolutionary relationship contain roughly the same conserved motifs, indicating that they may have similar functions (Figure 4b). All 19 *CbuSPL* proteins have a complete SBP conserved domain (Figure 4c). In the analysis of gene structure, the number of exons and introns contained in *CbuSPL* is very different. *CbuSPL17* has the largest number of exons and introns, with 11 and 10, respectively, while *CbuSPL4*, *CbuSPL5*, and *CbuSPL6* contain only 2 exons and 1 intron. In addition, a total of 12 *CbuSPL* genes have 5′-UTR and 3′-UTR and the remaining 7 members lack 5′-UTR and 3′-UTR. It can be found that *CbuSPL* genes in the same branch have similar gene structures (Table 1 and Figure 4d).

### 2.5. Chromosome Distribution and Synteny Analysis of the CbuSPL Genes in C. bungei

Gene duplication events are common and widely occur in plant gene family formation, which is important for understanding the adaptive evolution of species. To understand the duplication events of all *CbuSPL* genes in *C. bungei*, we performed synteny analysis using MCscanX and visualized with Advanced Circos in TBtools v1.127 [39,40,41]. We analyzed tandem duplication events between *CbuSPL* genes and found that there was only one gene pair tandem duplication event on chromosome 15 (Appendix A). Furthermore, we identified a total of six gene pairs with segmental duplication events, which occurred on 7 of the 20 chromosomes (Figure 5 and Appendix A), suggesting that segmental duplication plays an important role in the expansion of the *SPL* gene family in *C. bungei*. We also performed gene selection pressure analysis, using TBtools v1.127 to calculate the non-synonymous substitution (Ka) and synonymous substitution (Ks) values of the *SPL* gene family segmental duplication and tandem duplication gene pairs in *C. bungei*. It was found that the Ka/Ks ratio of each gene pair was less than 1, indicating that the *CbuSPL* gene family may have experienced strong purification selection pressure during evolution (Appendix A).

### 2.6. Syntenic Relationships of SPL Genes between C. bungei and Other Species

To further explore the synteny relationships between *CbuSPL* genes and related genes from other six representative species, including five eudicots (*Arabidopsis thaliana*, *Populus trichocarpa*, *Malus domestica*, *Sesamum indicum*, and *Solanum lycopersicum*) and one monocot (*Oryza sativa*), we performed comparative synteny analysis. The numbers of orthologous gene pairs were 15 between *C. bungei* and Arabidopsis, 38 between *C. bungei* and poplar, 44 between *C. bungei* and apple, 29 between *C. bungei* and sesame, 19 between *C. bungei* and tomato, and 5 between *C. bungei* and rice (Figure 6). It can be seen that there are few gene pairs between *C. bungei* and rice, which may be due to the closer phylogenetic relationship between dicots than that between monocots.

### 2.7. Analysis of Cis-Acting Elements in CbuSPL Promoters

The promoter sequence located upstream of the gene coding sequence is distributed with many cis-acting elements for regulating gene-specific expression. In order to better understand the potential regulatory mechanisms of *CbuSPL* genes in growth and development, plant hormone response, and stress response in *Catalpa bungei*, we further analyzed the 2000 bp promoter sequences upstream of *CbuSPL* genes and found a total of 47 types of cis-acting elements, including 22 light-responsive, 11 phytohormone-responsive, 8 plant growth-related, and 6 stress-responsive elements, respectively (Figure 7 and Appendix A). Among these cis-acting elements, light-responsive regulatory elements accounted for the largest proportion, including G-Box and Box 4, and others were distributed in most *CbuSPL* promoter regions. In addition, phytohormone regulatory elements were identified in most of the *CbuSPL* promoters, among which TGA-element, ABRE, P-box, and TCA-element were involved in the auxin response, abscisic acid response, gibberellin response, and salicylic acid response, respectively, while CGTCA-motif and TGACG-motif were involved in the methyl jasmonate (MeJA) response. Among plant-growth-related elements, a large number of CAT-box elements related to meristem expression were detected, and circadian elements were also found on the promoter regions of *CbuSPL5*, *CbuSPL7*, and *CbuSPL9*. Regarding stress response, ARE elements related to anaerobic induction, TC-rich repeats elements related to defense and stress responses, and LTR elements related to low-temperature response were more prevalent. The above results indicate that *CbuSPL* genes have potential roles in photosynthesis, abscisic acid, MeJA, circadian rhythm, meristem expression, and stress response.

### 2.8. Tissue-Specific Expression Analysis of CbuSPL Genes in C. bungei

In order to explore the expression pattern of *CbuSPL* genes in different tissues of *C*. *bungei*, we analyzed the expression of *CbuSPL* genes in different tissues from flower buds, leaves, petioles, and stems of *C. bungei* based on the previous RNA-seq data in the laboratory (Appendix A) and used TBtools v1.127 to draw a cartoon heat map. The results showed that the expression levels of *CbuSPL3*, *CbuSPL4*, *CbuSPL5*, *CbuSPL6*, *CbuSPL8*, *CbuSPL9*, *CbuSPL11*, *CbuSPL17*, and *CbuSPL18* genes were significantly higher in flower buds, while the expression levels of *CbuSPL2*, *CbuSPL12*, and *CbuSPL19* genes were higher in leaves. The expression levels of all 19 *CbuSPL* genes in petioles were lower. In addition, except for *CbuSPL2*, *CbuSPL4*, *CbuSPL5*, *CbuSPL8*, *CbuSPL9*, *CbuSPL11*, *CbuSPL12*, and *CbuSPL19*, the expression levels of the other genes in stems were relatively high (Figure 8). The above results indicate that the function of *CbuSPL* genes has differentiated and may be involved in flowering development, flowering transition, leaf development, stem development, and secondary growth in *C. bungei*.

### 2.9. Expression Pattern Analysis of CbuSPL Genes in Flower Buds of C. bungei and Catalpa ‘Bairihua’ at Different Developmental Stages

In order to explore the function of *CbuSPL* genes in the flowering transition and flowering development of *Catalpa* ‘Bairihua’, we analyzed the expression pattern of *CbuSPL* genes in the flower buds of *C. bungei* (normal flowering) and *Catalpa* ‘Bairihua’ (early flowering) during vegetative period (Vp), transition period (Tp), and reproductive period (Rp). Using the previous RNA-seq data from the laboratory [42], the heat map analysis was carried out (Appendix A). The results showed that except for *CbuSPL16*, *CbuSPL7*, *CbuSPL14*, and *CbuSPL17*, the expression levels of all other genes in EF vegetative period were significantly higher than those in the same period of NF, indicating that the *CbuSPL* gene family plays an important role in the flowering initiation. The expression levels of these four genes during the EF transition period were higher than those at the same period as NF, suggesting that these four genes may play a role in the flowering transition process. In addition, *CbuSPL2*, *CbuSPL18*, *CbuSPL9*, *CbuSPL5*, *CbuSPL12*, *CbuSPL6*, *CbuSPL10*, *CbuSPL4*, and *CbuSPL13* genes have higher expression levels in the EF reproductive period, indicating that these genes may play a specific function when the flower buds of *Catalpa* ‘Bairihua’ are in the reproductive growth stage (Figure 9).

### 2.10. Ectopic Expression of CbuSPL4 in Arabidopsis

To explore the function of *CbuSPL4* in flower development, we transformed *CbuSPL4* driven by the cauliflower mosaic virus 35S (CaMV 35S) promoter into wild-type Arabidopsis (*35S::CbuSPL4*). It was found that the transgenic plants showed obvious morphological changes compared with the wild type, including flowering time, plant height, leaf number, and leaf size (Figure 10a,b). The ectopic expression of the *CbuSPL4* gene in wild-type Arabidopsis advanced the phase transition, accelerated the transition from juvenile to the adult phase, and led to early flowering under LD conditions, indicating that *CbuSPL4* is roughly equivalent to its homologous gene *AthSPL3/4/5* in *A. thaliana* [43] (Appendix A). We next examined the expression of flowering-related genes in the inflorescences of *35S::CbuSPL4* transgenic plants. As expected, the expression levels of *AGL24*, *LFY*, *SOC1/AGL20*, *FT*, *FUL/AGL8*, and *AP1/AGL7* were significantly induced in transgenic plants compared with WT (Figure 10c).

## 3. Discussion

*Catalpa bungei* is an ancient tree species with ornamental, economic, and medicinal values unique to China [21,22,23]. It takes a long time to flower; generally, it takes 5–7 years for it to blossom, which seriously limits the work of hybrid breeding and genetic improvement [20,24]. *Catalpa* ‘Bairihua’ is an excellent hybrid with multi-season flowering obtained through the hybrid breeding of *Catalpa bungei* ‘Luo Qiu 4′ and *C. fargesii* Bur. f. *duclouxii* (http://rif.caf.ac.cn/News.aspx?ItemID=6009, accessed on 20 June 2022) [25,26]. Moreover, it can bloom in the same year through asexual reproduction such as grafting, which is very rare in woody plants [25]. Therefore, it is crucial to explore the molecular mechanisms of flowering time regulation and the reproductive transition of *Catalpa* ‘Bairihua’ for its genetic improvement and utilization. As a plant-specific transcription factor, the *SPL* gene family plays a critical role in plant growth and development. Although the *SPL* gene family has been identified in diverse plant species, there have been no genome-wide identification or systematic study reports on the *SPL* gene family in *Catalpa bungei*. In this study, we identified 19 putative *SPL* gene family members in the *C. bungei* genome, which is consistent with the number of *SPL* genes identified in sesame (Figure 3), which is similar to the homologous relationship of *C. bungei*. In addition, the amino acid number of these 19 CbuSPL proteins ranged from 170 to 1088, which was basically consistent with the number of Arabidopsis SPL proteins [27].

Through the tissue-specific expression analysis of *CbuSPL* genes, it was found that there were significant differences in the expression levels of different genes in flower buds, leaves, petioles, and stems (Figure 8), indicating that the function of the *CbuSPL* genes has undergone tissue differentiation, which may be involved in the flowering development, flowering transition, leaf development, stem development, and secondary growth of *C. bungei*. In addition, we explored the expression patterns of the *CbuSPL* genes in flower buds during the vegetative period (Vp), transition period (Tp), and reproductive period (Rp) of *C. bungei* (normal flowering) and *Catalpa* ‘Bairihua’ (early flowering). It was found that specific genes were highly expressed at different developmental stages (EF-Vp, EF-Tp, and EF-Rp) of *Catalpa* ‘Bairihua’. For example, *CbuSPL16*, *CbuSPL7*, *CbuSPL14*, and *CbuSPL17* genes were significantly higher in the EF transition period (EF-Tp), which may be related to the flowering transition. However, the expression of *CbuSPL* genes fluctuated in the flower buds at different developmental stages (NF-Vp, NF-Tp, and NF-Rp) of *C. bungei*. That is to say, although *C. bungei* did not bloom in the same year, in fact, the expression of *CbuSPL* genes was different at different stages of vegetative growth, and the genes that may control flowering development and leaf development were differentially expressed (Figure 9). In summary, this is consistent with previous reports that the *SPL* gene is involved in flowering initiation [7,8], phase transition from juvenile to adult, vegetative growth to reproductive growth [9,10,11], floral organ development [12], and fruit development [13].

The *CbuSPL4* gene was ectopically expressed in Arabidopsis to explore its function. The results showed that the transgenic plants had significant differences in flowering time, plant height, leaf number, and size compared with the wild type (Figure 10a,b). By analyzing the best homologous hits of *CbuSPL* genes in *Arabidopsis thaliana*, it was found that *CbuSPL4* and *AthSPL3/4/5* were classified into one cluster (Appendix A). This study also confirmed that the function of *CbuSPL4* is roughly equivalent to that of *AthSPL3/4/5* and can promote flowering [43]. AGL24 encodes a MADS-box protein involved in flowering, regulates the expression of SOC1, and is also upregulated by SOC1. AGL20/SOC1 controls flowering and is required for CO to promote flowering. AGL20/SOC1 acts with AGL24 to promote flowering and inflorescence meristem identity. LFY encodes a transcriptional regulator that promotes flowering transition and is involved in floral meristem development. FT together with LFY promotes flowering. FUL/AGL8 overexpression flowered early under SD and LD conditions and was negatively regulated by APETALA1 [44,45]. AP1 specifies the identity of floral meristem and sepals. The overexpression of AP1 promotes flowering, and there is a transition from apical and lateral branches to flowers [46]. We examined the expression of *AGL24*, *LFY*, *SOC1/AGL20*, FT, *FUL/AGL8*, and *AP1/AGL7* genes in *35S::CbuSPL4* transgenic plants and found that the expression levels of these positively regulated flowering genes were significantly induced. In the future, we will further verify its gene function in *C. bungei* and conduct research on the regulatory relationship between *CbuSPL4* and these genes to improve its regulatory network.

## 4. Materials and Methods

### 4.1. Plant Materials and Growth Conditions

*Catalpa bungei* is a perennial woody plant. *Catalpa* ‘Bairihua’ is an excellent hybrid with multi-season flowering obtained through the hybrid breeding of *Catalpa bungei* ‘Luo Qiu 4′ and *C. fargesii* Bur. f. *duclouxii* (http://rif.caf.ac.cn/News.aspx?ItemID=6009, accessed on 20 June 2022) [25]. Moreover, it can bloom in the same year through asexual reproduction such as grafting, which is very rare in woody plants. *Catalpa* ‘Bairihua’ plants under standard water management and pest control were grown via grafting in the *Catalpa* test forest base of Luoyang City, Henan Province, China. Flower bud samples at different developmental stages were collected from 24 January to 24 March 2021 for expression analysis and transcriptome sequencing. The Arabidopsis were grown in the greenhouse under 16 h/8 h, light/dark at 22 °C. All samples were collected in the morning in 5 mL cryovials and immediately stored in liquid nitrogen.

### 4.2. Identification of SPL Gene Family in C. bungei

To identify the *SPL* gene family in the *C. bungei* genome, the amino acid sequences of 17 known *SPL* family genes [27] in *Arabidopsis* obtained from The Arabidopsis Information Resource (TAIR) (https://www.arabidopsis.org/, accessed on 6 July 2022) [47] were used as query sequences to perform a BLASTP search against the protein sequences of *C. bungei* data that annotated according to the *C. bungei* genome (the entire *C. bungei* genome was sequenced by our research group, and the related paper is in preparation) using a cutoff e-value of 1 × 10^−5^. After this initial screening, the HMM model file SBP domain (PF03110) was downloaded from the Pfam 35.0 database (http://pfam.xfam.org/, accessed on 6 July 2022) [48], and we searched the genome protein databases with an e-value cutoff of 1 × 10^−5^ using HMMER v3.3.2 software [49]. Subsequently, candidate *SPL* family genes and were then verified with the Batch CD-Search Tool (https://www.ncbi.nlm.nih.gov/Structure/bwrpsb/bwrpsb.cgi, accessed on 6 July 2022) [50], Pfam (http://pfam.xfam.org/, accessed on 6 June 2022) [48], and SMART (a Simple Modular Architecture Research Tool) (http://smart.embl.de/, accessed on 6 July 2022) [51] to ensure the completeness of the SBP domain. Redundant sequences or sequences with an incomplete SBP domain were excluded from the following analyses. Finally, each *SPL* gene was named according to the position on the chromosome from top to bottom. Molecular weight (Mw) and the theoretical isoelectric point (pI) of the *SPL* family genes in *C. bungei* were determined using the Compute pI/Mw tool (https://web.expasy.org/compute_pi/, accessed on 10 July 2022) of ExPASy (https://www.expasy.org/, accessed on 10 July 2022) [52,53]. Subcellular localization prediction was performed using DeepLoc-2.0 (https://services.healthtech.dtu.dk/services/DeepLoc-2.0/, accessed on 10 July 2022) [54].

### 4.3. Multiple-Sequence Alignment and Visualization of SBP Domain

Multiple-sequence alignment of SBP domains of CbuSPL proteins was performed using the Muscle program in MEGA11 v11.0.13 [55], and the results were visualized using Jalview v2.11.2.0 [56]. Sequence logos of SBP domains were drawn with TBtools v1.127 [40].

### 4.4. Phylogenetic Analysis

To infer the evolutionary history of *CbuSPL* genes, we selected six species, including five eudicots (*Arabidopsis thaliana*, *Populus trichocarpa*, *Malus domestica*, *Sesamum indicum*, and *Solanum lycopersicum*) and one monocot (*Oryza sativa*). Genome sequences and annotation files of Arabidopsis (Araport11) [57], *Populus trichocarpa* (v4.1) [58], apple (v1.1) [59], tomato (ITAG4.0) [60], and rice (v7.0) [61] were downloaded from Phytozome v13 (https://phytozome-next.jgi.doe.gov/, accessed on 20 October 2022) [62], while that of sesame (v1.0) [63] was downloaded from Ensembl Plants (http://plants.ensembl.org/index.html, accessed on 20 October 2022). Using the method mentioned above, the *SPL* gene family members from these species were identified for phylogenetic tree construction.

Multiple sequence alignments of *SPL* genes from *C. bungei* and 6 other species were performed using the Muscle program in MEGA11 [55], and then we used it to create maximum-likelihood phylogenetic trees with 5000 ultrafast bootstrap replicates using IQ-TREE v1.6.12 [64,65]. The tree was visualized using iTOL v 6.7.1 (https://itol.embl.de/, accessed on 10 March 2023) [66].

### 4.5. Conserved Motifs and Gene Structure Analysis

The online tool MEME v 5.5.1 (https://meme-suite.org/meme/tools/meme, accessed on 20 March 2023) was used to identify the conserved motifs of *CbuSPL* genes; the number of motifs was set to 10 and other parameters were default [67]. Based on the *C. bungei* genome annotation files, the gene structure was analyzed using TBtools v1.127, and the phylogenetic tree, conserved motifs, conserved domains, and gene structure were integrated and visualized using TBtools v1.127 [40].

### 4.6. Syntenic Analysis between C. bungei and Other Species

Syntenic analysis between *C. bungei* and other 6 species was performed by MCScanX program of TBtools v1.127 [39,40]. Using genome sequences and annotation files (gff3 or gtf format) as input data, the syntenic blocks for each pair of species were identified, with default parameters.

### 4.7. Cis-Elements Analysis

The 2000 bp upstream sequence of CDS of *CbuSPL* genes in *C. bungei* genome was obtained using TBtools v1.127 [40]. Then, the cis-elements of *CbuSPL* promoters were screened using the PlantCARE website (https://bioinformatics.psb.ugent.be/webtools/plantcare/html/, accessed on 10 July 2023) [68].

### 4.8. Expression Pattern Analysis

In order to explore the expression pattern of *CbuSPL* genes, we used the previous RNA-seq data to analyze the expression of *CbuSPL* genes in different tissues including flower buds, leaves, petioles, and stems, and explored the expression patterns of *CbuSPL* genes in flower buds during the vegetative period (Vp), transition period (Tp), and reproductive period (Rp) of *C. bungei* (normal flowering) and *Catalpa* ‘Bairihua’ (early flowering). The original data are shown in Appendix A.

### 4.9. Ectopic Expression of CbuSPL4 in Arabidopsis

The full-length coding sequences of *CbuSPL4* were cloned into pENTR/D-TOPO vector (Invitrogen, Waltham, MA, USA) after sequencing confirmation and inserted into the pBI121 binary vector [69] under the control of the CaMV 35S promoter with the In-Fusion HD Cloning Kit (TaKaRa, Dalian, China). The recombinant construct was introduced into *Agrobacterium tumefaciens* strain GV3101 and then transformed into wild-type Arabidopsis by the floral dip method [70,71]. The transgenic plants were screened on Murashige and Skoog (MS) medium [72] with 50 mg·mL^−1^ kanamycin until homozygous lines were obtained. The primers used for vector construction are given in Appendix A.

### 4.10. Total RNA Extraction and RT-qPCR

The total RNA from transgenic and wild-type Arabidopsis was extracted using an RNeasy Plant Mini Kit (QIAGEN, Hilden, Germany). RNA quantity and purity were assessed with a NanoDrop spectrophotometer (Thermo Fisher Scientific, Waltham, MA, USA).

Reverse-transcription quantitative PCR (RT-qPCR) was performed to detect the expression of flowering-related genes in transgenic and wild-type Arabidopsis. One microgram of total RNA from each sample was used for reverse transcription to generate cDNA using the PrimeScript RT reagent Kit with gDNA Eraser (TaKaRa) according to the manufacturer’s protocol. RT-qPCR was performed using TB Green Premix Ex Taq II (Tli RNaseH Plus) (TaKaRa) on a 7500 Fast Real-Time PCR System machine (Applied Biosystems, Waltham, MA, USA). Three biological and three technical replicates were performed, and the relative expression of the target gene was calculated with the 2^−ΔΔCT^ method [73]. *β*-*TUBULIN*-*2* (AT5G62690.1) was used as an internal reference gene [7]. The primers used for RT-qPCR are listed in Appendix A.

### 4.11. Heatmap Analysis

The heatmap analysis of cis-elements, tissue-specific expression, and developmental stage-specific expression was carried out using TBtools v1.127 [40]. The entire raw data are provided in the Appendix A.

### 4.12. Statistical Analysis

All statistical analysis and graphing were performed using Excel and GraphPad Prism v9.0.0 (121).

## 5. Conclusions

To the best of our knowledge, this is the first report on the genome-wide analysis of the *SPL* genes in *Catalpa bungei*. A total of 19 *CbuSPLs* were identified in this study, all of which have a complete SBP domain. According to the phylogenetic relationship, they can be divided into eight groups, and the genes in the same group have similar gene structure and conserved motifs. Synteny analysis showed that *CbuSPL* genes were unevenly distributed on chromosomes, and also suggested that fragment duplication played an important role in the expansion of the *CbuSPL* gene family. At the same time, *CbuSPL* genes have cis-acting elements and functions related to light response, hormone response, growth and development, and stress response. Tissue-specific expression and developmental period-specific expression analysis showed that *CbuSPL* may be involved in flowering initiation and development, flowering transition, leaf development, and other processes. In addition, the ectopic expression of *CbuSPL4* in Arabidopsis confirmed that it can promote early flowering and induce the expression of related flowering genes.

## Figures and Tables

**Figure 1 ijms-25-00097-f001:**
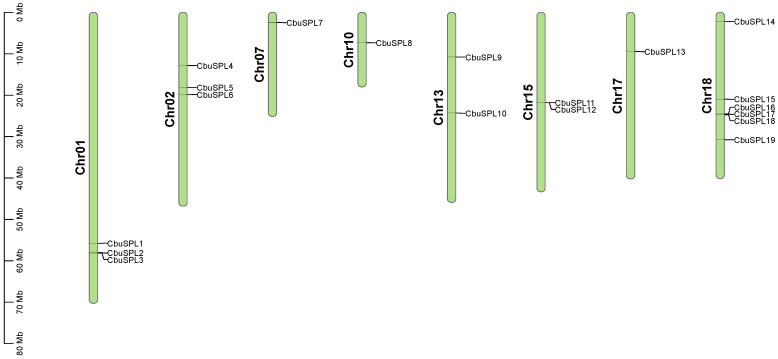
Distribution of *CbuSPL* genes on the *C. bungei* chromosomes.

**Figure 2 ijms-25-00097-f002:**
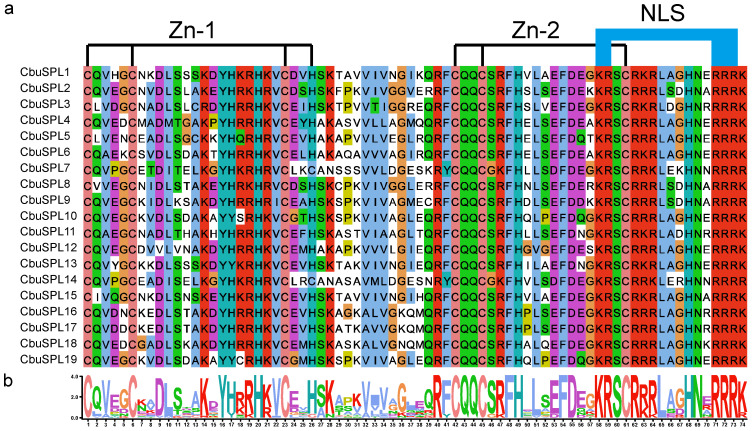
Multiple sequence alignment of SBP domains from CbuSPL proteins. (**a**) Multiple alignment of SBP domains; (**b**) sequence logo of the SBP domains.

**Figure 3 ijms-25-00097-f003:**
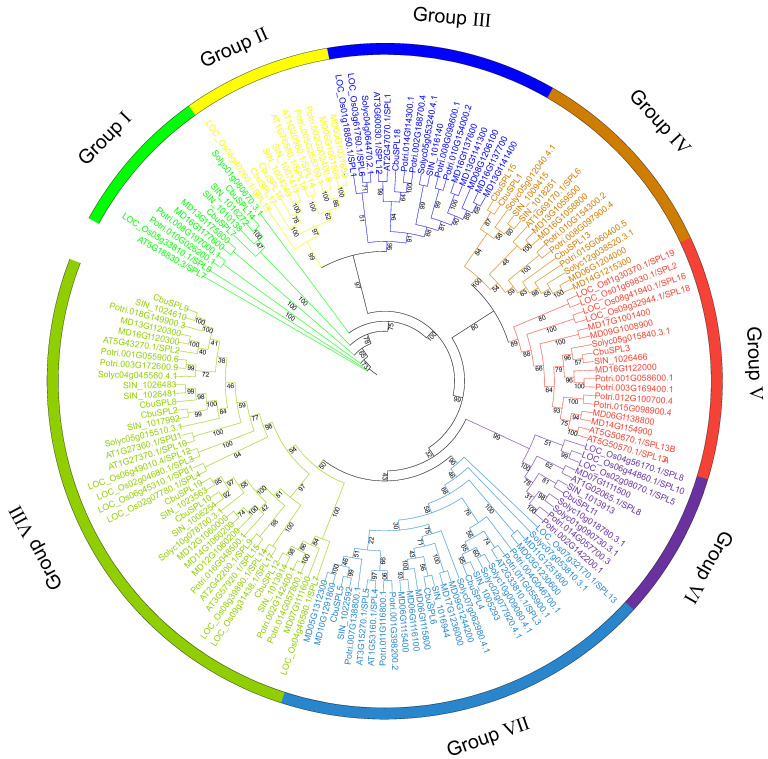
Maximum-likelihood phylogenetic tree of SPL family proteins in *C. bungei*, *Arabidopsis thaliana*, *Populus trichocarpa*, *Malus domestica*, *Sesamum indicum*, *Solanum lycopersicum*, and *Oryza sativa*. Branches and labels of different colors represent different groups, and the numbers at nodes represent bootstrap values.

**Figure 4 ijms-25-00097-f004:**
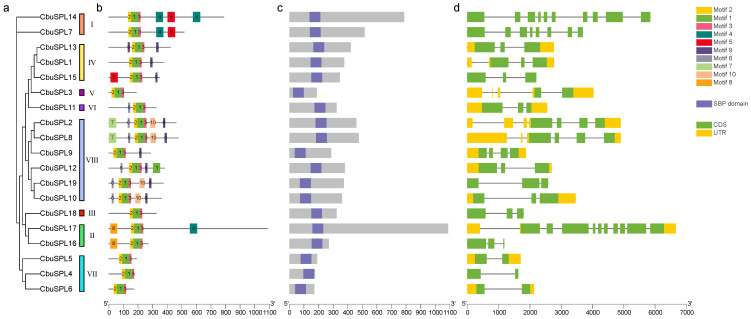
Phylogenetic relationship, conserved motif, and gene structure analysis of *CbuSPL* genes in *C. bungei*. (**a**) Phylogenetic tree of all CbuSPL proteins constructed using maximum-likelihood method; (**b**) motif distribution of CbuSPL proteins; Motifs 1–10 are shown as rectangular boxes of different colors; (**c**) conserved domain distribution of CbuSPL proteins; the purple box indicates the SBP domain in the corresponding amino acid sequence; (**d**) gene structures of *CbuSPL* genes arranged according to phylogenetic relationship; yellow boxes represent 5′ UTR and 3′ UTR, green boxes represent exons, and gray lines represent introns.

**Figure 5 ijms-25-00097-f005:**
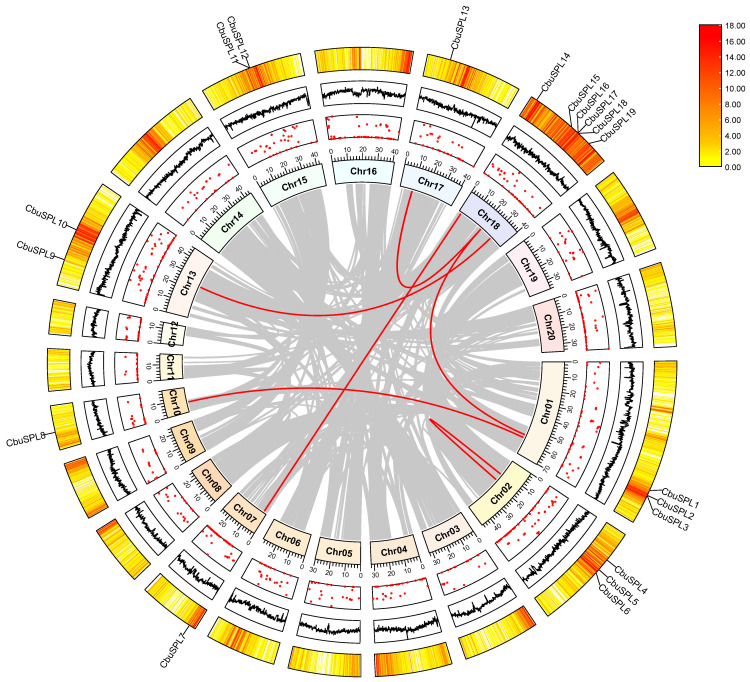
Collinearity analysis of the *SPL* gene family in *C. bungei*. From the inside to the outside, the gradient color rectangle represents chromosomes 01–20, the red dot represents the gap distribution on the genome, the black line represents the GC ratio on the genome, and the heat map of the outermost circle represents the gene density. In the center, the gray lines indicate synteny blocks in the *C. bungei* genome, while the red lines between chromosomes delineate segmental duplication gene pairs.

**Figure 6 ijms-25-00097-f006:**
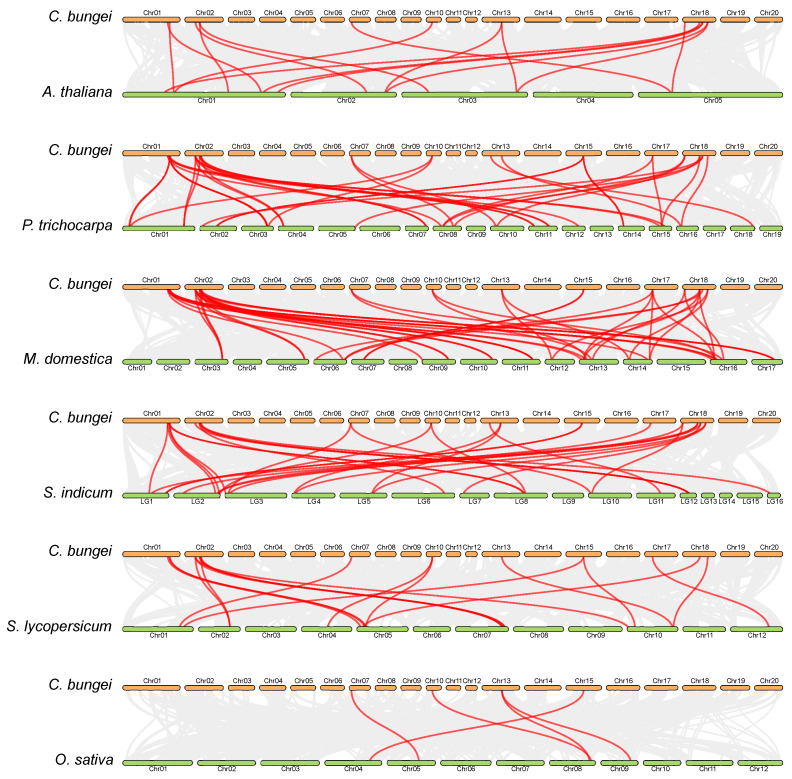
Synteny analysis of the *CbuSPL* genes between *C. bungei* and six other plant species. The gray lines indicate gene blocks in *C. bungei* that are orthologous to the other genomes. The red lines delineate the syntenic *SPL* gene pairs.

**Figure 7 ijms-25-00097-f007:**
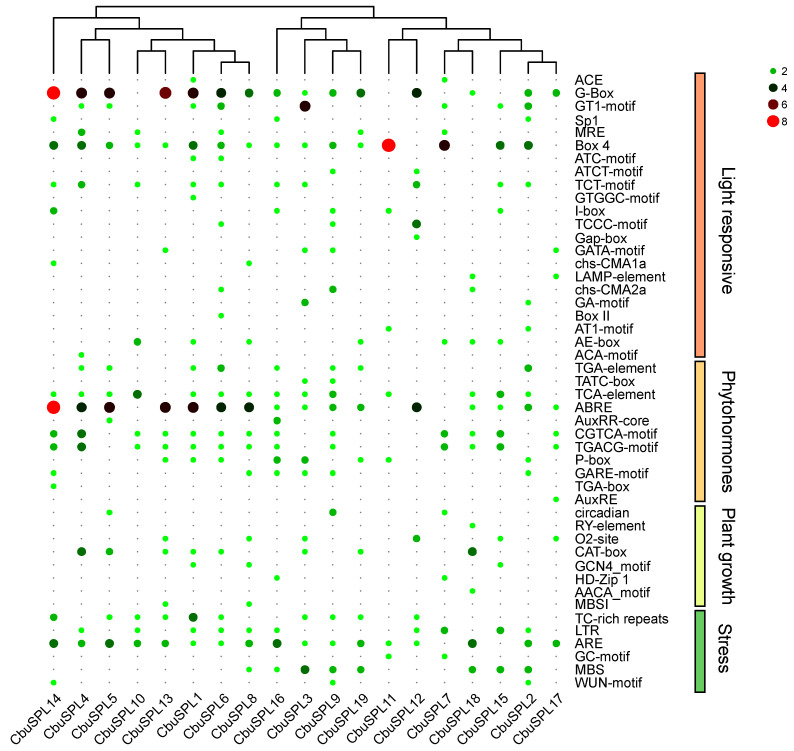
Analysis of cis-acting elements in the promoter region of the *CbuSPL* genes. Based on functional annotation, *CbuSPL* gene cis-acting elements can be classified into four categories: light-responsive, phytohormone-responsive, plant-growth-related, and stress-responsive.

**Figure 8 ijms-25-00097-f008:**
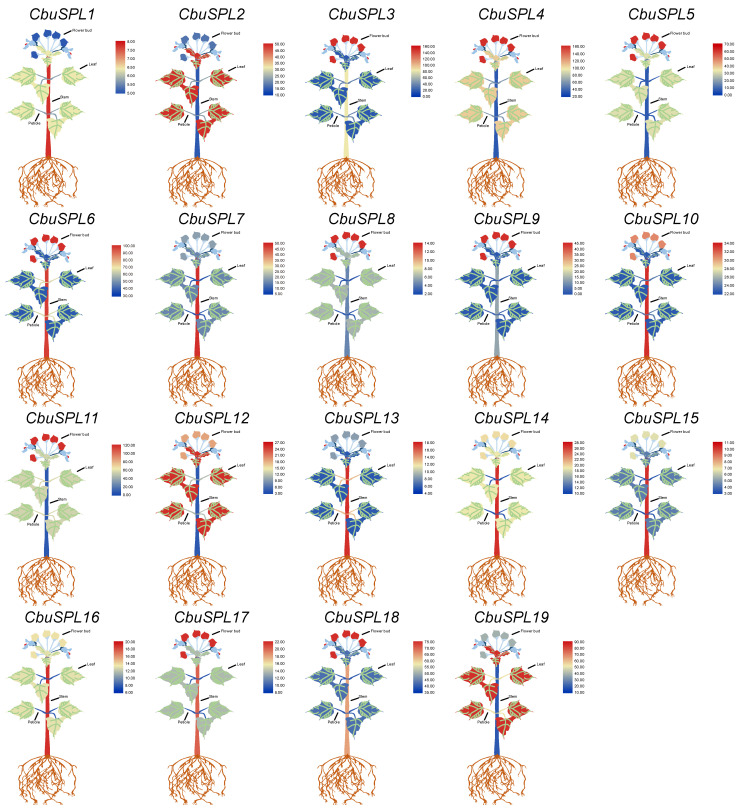
Expression analysis of *CbuSPL* genes in flower buds, leaves, petioles, and stems of *C. bungei*. Expression values are based on RNA-seq, visualized using TBtools v1.127. Red represents high expression levels and blue represents low expression levels.

**Figure 9 ijms-25-00097-f009:**
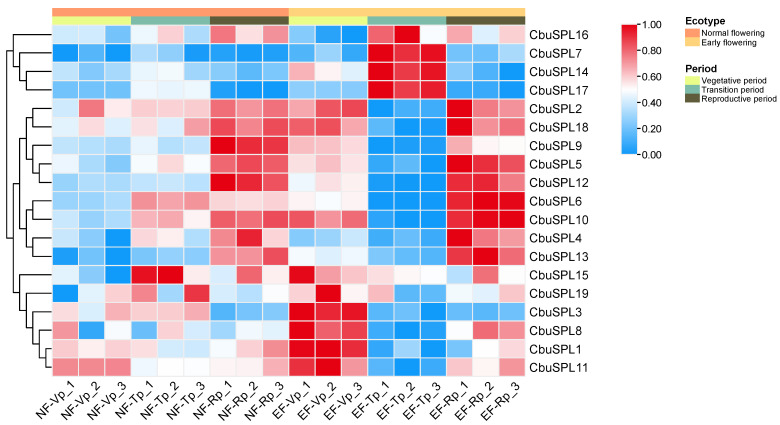
Expression pattern analysis of *CbuSPL* genes in flower buds of *C. bungei* and *Catalpa* ‘Bairihua’ at different developmental stages. NF: Normal flowering (*C. bungei*), EF: Early flowering (*Catalpa* ‘Bairihua’); VP: Vegetative period, TP: Transition period, RP: Reproductive period.

**Figure 10 ijms-25-00097-f010:**
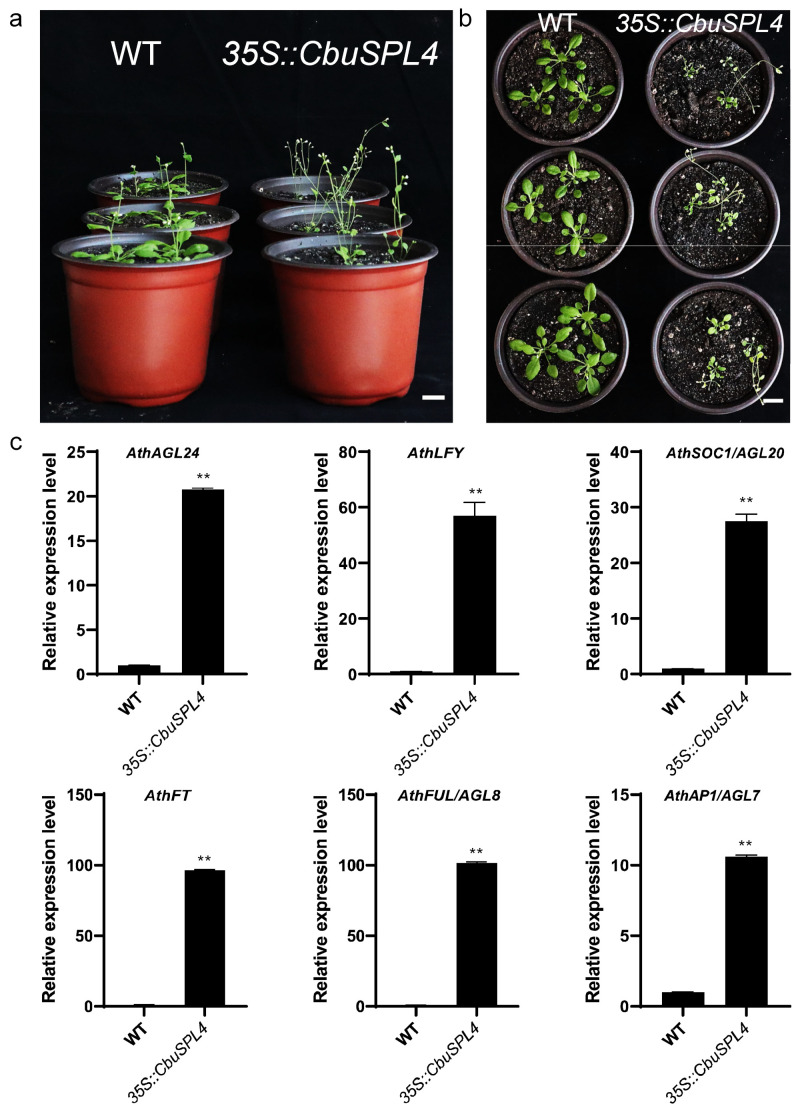
Ectopic expression of *CbuSPL4* in Arabidopsis. (**a**,**b**) Phenotypic characterization of *35S::CbuSPL4* transgenic plants. (**c**) Expression analysis of flowering-related genes in transgenic Arabidopsis and wild-type plants through real time RT-PCR. Error bars represent SE values of three biological replicates. Asterisks indicate significant differences between transgenics and wild-type plants, determined with Student’s *t* test (**, *p* < 0.01). Bars = 2 cm.

**Table 1 ijms-25-00097-t001:** Characteristics properties of *CbuSPLs* in *C. bungei*.

Gene Name	Gene ID	Chromosomal Location	Number of Exon	Number of Intron	Gene Length (bp)	CDS Length (bp)	Number of Amino Acid (aa)	Molecular Weight (kDa)	Theoretical pI	Subcellular Localization
*CbuSPL1*	evm.TU.group0.1477	Chr01:55742624-55745394	4	3	2771	1128	375	41,716.90	8.65	Nucleus
*CbuSPL2*	evm.TU.group0.1766	Chr01:58009013-58013912	7	6	4900	1383	460	50,055.82	8.68	Nucleus
*CbuSPL3*	evm.TU.group0.1777	Chr01:58096915-58100943	5	4	4029	564	187	20,922.46	9.26	Nucleus
*CbuSPL4*	evm.TU.group1.669	Chr02:12828539-12830167	2	1	1629	528	175	20,058.44	8.18	Nucleus
*CbuSPL5*	evm.TU.group1.1223	Chr02:18112076-18113781	2	1	1706	570	189	21,333.84	8.60	Nucleus
*CbuSPL6*	evm.TU.group1.1470	Chr02:19850125-19852249	2	1	2125	513	170	18,894.06	9.38	Nucleus
*CbuSPL7*	evm.TU.group14.282	Chr07:2454793-2458481	8	7	3689	1548	515	57,276.22	8.16	Nucleus
*CbuSPL8*	evm.TU.group17.394	Chr10:7319282-7324180	6	5	4899	1431	476	51,933.40	8.56	Nucleus
*CbuSPL9*	evm.TU.group2.188	Chr13:10783205-10785073	4	3	1869	861	286	31,903.87	8.86	Nucleus
*CbuSPL10*	evm.TU.group2.1122	Chr13:24327368-24330828	3	2	3461	1083	360	38,249.37	9.40	Nucleus
*CbuSPL11*	evm.TU.group4.922	Chr15:21772905-21775454	3	2	2550	975	324	35,676.03	8.16	Nucleus
*CbuSPL12*	evm.TU.group4.923	Chr15:21785690-21788384	3	2	2695	1143	380	41,706.53	8.74	Nucleus
*CbuSPL13*	evm.TU.group6.274	Chr17:9460552-9463317	3	2	2766	1266	421	47,028.82	8.69	Nucleus
*CbuSPL14*	evm.TU.group7.217	Chr18:2162703-2168546	10	9	5844	2364	787	88,053.29	6.12	Nucleus
*CbuSPL15*	evm.TU.group7.2043	Chr18:20920539-20922742	3	2	2204	1041	346	38,610.53	6.80	Nucleus
*CbuSPL16*	evm.TU.group7.2477	Chr18:24516899-24518078	3	2	1180	813	270	30,105.91	9.59	Nucleus
*CbuSPL17*	evm.TU.group7.2481	Chr18:24528473-24535126	11	10	6654	3267	1088	119,687.73	6.78	Nucleus
*CbuSPL18*	evm.TU.group7.2509	Chr18:24721085-24722881	3	2	1797	975	324	36,178.42	9.31	Nucleus
*CbuSPL19*	evm.TU.group7.3182	Chr18:30778588-30781167	3	2	2580	1122	373	40,566.42	9.35	Nucleus

## Data Availability

Data are contained within the article and Appendix A.

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
