# Peer review of "Genome-Wide Identification and Expression Analysis of the SQUAMOSA Promoter-Binding Protein-like (SPL) Transcription Factor Family in Catalpa bungei"

_ijms, 2023, doi:10.3390/ijms25010097_

Round 1
Reviewer 1 Report
Comments and Suggestions for Authors
Congratulation!
The article is very clear, easy to read an interesting about a plant-specific transcription factor, impacting the growth and development in plant - Catalpa bungei in the study.
The authors identified putative SPL gene family members in C. 20 bungei genome, divided into 8 group. The analysis showed that fragment duplication played an important role in the expansion of the CbuSPL gene family and CbuSPL genes has functions related to hormone response, growth and development, and stress response. Tissue-specific expression and developmental period-specific expression analysis showed that CbuSPL may be involved in flowering initiation and development, flowering transition, leaf development.
The figure has high resolution and the results are clear, adequately discussed.
Author Response
Thank you for your congratulations!
Thank you very much for taking the time to review this manuscript.
Thank you very much for your approval of this manuscript!
Reviewer 2 Report
Comments and Suggestions for Authors
Dear Authors,
Reviewer comments ijms-2721593
The manuscript entitled „Genome-wide identification and expression analysis of the SQUAMOSA promoter-binding protein-like (SPL) transcription factor family in Catalpa bungei“ represents a useful study aimed at an investigation of a plant-specific SPL transcription factor family in Catalpa bungei. In the study, CbuSPL genes chromosomal localization, multiple sequence alignments, phylogenetic tree, conserved motifs analysis, collinearity analysis, synteny analysis, cis-elements promoter analysis, and expression analysis in C. bungei plant organs in C. bungei cv. Bairihua.
I can recommend the manuscript for publication in International Journal of Molecular Sciences.
I have only a few comments on the present manuscript which are given below:
1/ In Figure 3 legend, a brief description of the nodes at the branches of the phylogenetic tree has to be added, e.g., „the numbers at nodes represent bootstrap values per 1000 replicates…“
2/ In Figure 8 legend, it is written that the expression values of CbuSPL genes in C. bungei organs were determined by RNA-seq approach. Thus, appropriate information on RNA-seq procedure used to determine tissue-specific expression of CbuSPL genes has to be added to Materials and methods section.
3/ Line 462: Supplementary materials. A complete list of supplementary materials attached to the manuscript has to be provided here.
4/ Formal comments on the text related to English language and style:
Abstract, line 24: Modify the verb form „has“ to „have“ in the statement: „At the same time, CbuSPL genes have cis-acting elements and functions….“
Line 272: Correct the typing error „adult phase“ (not „adulte phase“).
Final recommendation: Accept after a minor revision.

Dear Authors,
Reviewer comments ijms-2721593
The manuscript entitled „Genome-wide identification and expression analysis of the SQUAMOSA promoter-binding protein-like (SPL) transcription factor family in Catalpa bungei“ represents a useful study aimed at an investigation of a plant-specific SPL transcription factor family in Catalpa bungei. In the study, CbuSPL genes chromosomal localization, multiple sequence alignments, phylogenetic tree, conserved motifs analysis, collinearity analysis, synteny analysis, cis-elements promoter analysis, and expression analysis in C. bungei plant organs in C. bungei cv. Bairihua.
I can recommend the manuscript for publication in International Journal of Molecular Sciences.
I have only a few comments on the present manuscript which are given below:
1/ In Figure 3 legend, a brief description of the nodes at the branches of the phylogenetic tree has to be added, e.g., „the numbers at nodes represent bootstrap values per 1000 replicates…“
2/ In Figure 8 legend, it is written that the expression values of CbuSPL genes in C. bungei organs were determined by RNA-seq approach. Thus, appropriate information on RNA-seq procedure used to determine tissue-specific expression of CbuSPL genes has to be added to Materials and methods section.
3/ Line 462: Supplementary materials. A complete list of supplementary materials attached to the manuscript has to be provided here.
4/ Formal comments on the text related to English language and style:
Abstract, line 24: Modify the verb form „has“ to „have“ in the statement: „At the same time, CbuSPL genes have cis-acting elements and functions….“
Line 272: Correct the typing error „adult phase“ (not „adulte phase“).
Final recommendation: Accept after a minor revision.
Author Response
Thank you very much for taking the time to review this manuscript. Please see the attachment for your detailed response.

Reviewer 3 Report
Comments and Suggestions for Authors
The manuscript “Genome-Wide Identification and Expression Analysis of the SQUAMOSA Promoter-Binding Protein-like (SPL) Transcription Factor Family in Catalpa bungei" aims to explore and investigate the physicochemical properties, phylogenetic relationships, chromosome localization gene, and protein-conserved domain structure of SPL gene family and explored the role in flowering. The manuscript is well-written and technically sound. The results support the conclusion.
· I have minor revisions for the authors to improve the quality of the manuscript.
· There is not enough reference in the Introduction section.
· Table 1; Add the LOC/NCBI accession number for gene ID, please add the genome
version and access date.
· Figure 4, increase the font size and image resolution
· Please review the manuscript carefully as there were multiple instances of words breaking due to different line breaks during the preparation of the draft.
Additional comments:
1. Do you consider the topic original or relevant in the field? Does it address a specific gap in the field?
Response; The research is original and relevant to the field.
2. What does it add to the subject area compared with other published material?
Response; Previously, SPL genes were not well characterized in Catalpa bungei, the experiment results from this study are useful for further investigation.
3. What specific improvements should the authors consider regarding the methodology?
Response; In the methodology section 4.1- 4.7, the procedure is common, most of the data is obtained from the public domain, and analyzed by bioinformatics or other specific software (s). However, in 4.8 Ectopic expressions of CbuSPL4 in the Arabidopsis section, the authors provided limited information only, they should improve this section with a detailed protocol including generation (T1, T2, and t3), selection criteria, positive plants, and total plants.
4. What further controls should be considered?
Response; N/A
Comments on the Quality of English LanguageModerate editing of English language required
Author Response

(The authors gave the same response as above.)
